# Harmonic Clarity: Audio Source Separation Techniques on Classical Music

**Rim El Filali**
Department of CS & Engineering (ACP)
Tsinghua University
2024280042
ai-l24@mails.tsinghua.edu.cn

**Ziyad Fawzy**
Department of CS & Engineering (ACP)
Tsinghua University
2024280038
fjz24@mails.tsinghua.edu.cn

**Zhengyang Zhu**
Music Artificial Intelligence and Music Information Technology
Central Conservatory of Music
P24134005
zhengyang_zhu@mail.ccom.edu.cn

## 1   Background

Hearing loss affects music perception, often causing quiet passages to become inaudible, instruments to be unidentifiable, lyrics difficult to hear, and pitch to distort. Current hearing aids struggle with complex compositions, especially classical music. This challenge is theoretically and practically important, advancing the performance of Music Source Separation (MSS) on classical music's compositional complexity, while promoting emotional well-being and social inclusion for the hearing-impaired.

Separating instruments is crucial for effectively rebalancing a music piece. Rebalancing then allows for the creation of personalized remixes, enhancing the listening experience. Our approach involves developing an end-to-end pipeline that separates, rebalances, and remixes classical music, designed to benefit hearing aid users in both live and recorded settings. This system aims to broaden accessibility, creating a tailored auditory experience for users with hearing loss.

## 2   Related Work

Over the past decade, Audio Source Separation (ASS) gained considerable attention with various methods emerging such as local Gaussian modeling [1], non-negative factorization [2], and their combinations [3]. Techniques using Deep Neural Networks (DNNs) advanced separation performance beyond traditional methods. In [4], a standard feed-forward fully connected network (FNN) derived source spectra by aggregating temporal contexts across multiple frames. Building on this, [5] introduced an extended DenseNet architecture with up-sampling layers, block skip connections, and band-specific dense blocks, improving contextual learning and achieving state-of-the-art signal-to-distortion ratios on SiSEC 2016, while reducing parameter requirements and training time.

Music Source Separation (MSS) aims to isolate individual sound sources from a mixture, closely mirroring ASS but with key differences. MSS is essential in audio processing and speech enhancement and separation tasks [6, 7]. The higher sample rates and professional manipulation often found in music signals (e.g., 44.1k Hz) make MSS more challenging than narrow-band or wide-band speech enhancement and separation tasks. Thus, high-quality MSS models are critical for developing robust separation systems for complex real-world scenarios. Recent research has explored time-domain systems and the fusion of time-domain and frequency-domain systems. Wave-U-Net [8] adapted

38th Conference on Neural Information Processing Systems (NeurIPS 2024).

U-Net to a one-dimensional time domain for end-to-end separation, showing performance comparable to spectrogram-based models for singing voice isolation. Conv-TasNet [9] is a fully convolutional time-domain network using a linear encoder-decoder paired with a Temporal Convolutional Network (TCN), outperforming both traditional time-frequency masking methods in separating two and three-speaker mixtures and ideal time-frequency magnitude masks in two-speaker separation.

Despite the success of these methods, understanding their effectiveness on music data and how to adapt them to better capture music-specific characteristics remains challenging. BSRNN [10], a frequency-domain model, was proposed to tackle this by splitting spectrograms into sub-bands for interleaved modeling with bandwidth choices based on knowledge of target source. A semi-supervised fine-tuning method was also introduced to use unlabeled data. Results showed that BSRNN outperformed leading models in MDX Challenge 2021 and semi-supervised fine-tuning improved efficiency on all instrument tracks [10]. However, research on source separation in string quartets is limited, particularly in distinguishing between similar instruments such as two violins. The 2nd Cadenza Challenge[1] aims to address these challenges by focusing on the rebalancing of instrument levels within small classical music ensembles.

## 3  Proposed Method

We base our model on Mamba, a state space model architecture known for its efficiency and effectiveness in long-sequence modeling, specifically because it consistently outperforms other recent models in music source separation tasks [11, 12]. Mamba's efficient handling of temporal dependencies and its linear computational complexity make it an excellent candidate for tasks with complex, overlapping audio signals, such as music separation.

A core strength of Mamba is its bidirectional processing, enabling it to consider both past and future information within a sequence—a feature particularly valuable for isolating specific instruments in mixed audio. Additionally, Mamba's adaptive state update mechanism allows the model to prioritize and distinguish different musical features effectively, even when sources have overlapping spectral characteristics. This results in superior performance for music separation across varied conditions, supporting its selection in our approach [12]. In our experimental setup, we will explore two training configurations:

1. **Specialized Instrument Models**: Training separate Mamba-based models for each instrument (e.g., vocals, drums, bass) to optimize performance for isolating individual sources.
2. **Combined Instrument Model**: Training a single, unified model to handle the separation of all instruments, allowing the network to generalize across various musical elements.

By testing these configurations, we aim to assess the versatility of Mamba's performance across specialized and generalized tasks, optimizing our method based on the findings.

## 4  Evaluation

To evaluate the effectiveness of our approach, we will use the Hearing Aid Audio Quality Index (HAAQI) [13] as an objective metric for assessing audio quality. HAAQI is an intrusive metric that compares the remixed audio to a reference signal derived from the original isolated sources, which have been rebalanced according to the specified target gains. The average validation set HAAQI score is calculated as the mean of all average left and right scores. HAAQI is formulated as:

$$\text{HAAQI} = \frac{1}{nsamples} \sum_{i=1}^{nsamples} \frac{leftscore_i + rightscore_i}{2}, \tag{1}$$

where $leftscore_i$ is the HAAQI score for the left ear, $rightscore_i$ is the HAAQI score for the right ear and $nsamples$ is the total number of samples to evaluate. Moreover, we plan to conduct perceptual tests with individuals experiencing hearing loss to gather qualitative feedback and ensure that our adjustments improve their listening experiences.

---

[1]https://cadenzachallenge.org/docs/cadenza2/intro

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
