# OpenReview forum: "Harmonic Clarity: Audio Source Separation  Techniques on Classical Music"
_tsinghua.edu.cn/THU/2024/Fall/AML — THU 2024 Fall AML Submission_

### Official Review · ~Sui_Yuanpei1 · 2024-11-10
**Strong concept with potential for impactful applications, though limited practical testing for hearing-impaired users might reduce real-world relevance**

**Rating:** 8
**Confidence:** 5

**Review:**

This proposal presents an innovative approach to Music Source Separation (MSS) specifically for classical music, focusing on improving accessibility for hearing aid users. By utilizing the Mamba model for efficient, adaptive separation, the project addresses both the technical and social aspects of enabling hearing-impaired individuals to enjoy complex compositions. Given the compositional intricacies of classical music, this MSS work is especially significant in promoting inclusivity and enhancing auditory experiences for those with hearing impairments.

Pros:
1.The focus on MSS for classical music, targeting accessibility for hearing-impaired users, adds a valuable social dimension to the technical work.
2.The use of Mamba, known for its efficiency in long-sequence audio tasks, is well-suited for complex music separation and can address overlapping frequency challenges.
3.The proposal’s plan to experiment with both specialized and combined instrument models provides a robust methodology for evaluating MSS performance.
4.By combining HAAQI as an objective metric with perceptual testing for hearing-impaired listeners, the evaluation plan is thorough.

Cons:
1.There is a lack of emphasis on practical, large-scale testing with hearing-impaired users, which may impact the real-world applicability of the findings.
2.The high complexity of separating classical compositions, especially distinguishing similar instruments (e.g., multiple stringed instruments), may lead to performance limitations.
3.The success of the project is heavily dependent on Mamba’s handling of temporal dependencies in overlapping frequencies, which may require significant tuning.
4.Training models specific to individual instruments could limit generalizability across diverse musical compositions.

---

### Official Review · ~Matteo_Jiahao_Chen1 · 2024-11-10
**Good proposal for Music Source Separation in classical music**

**Rating:** 10
**Confidence:** 5

**Review:**

This work tackles the challenge of Music Source Separation (MSS) in classical music, aimed at enhancing the listening experience for hearing-impaired users. It introduces an approach based on  Mamba.
### Strengths
1. The use of the Mamba model provides an efficient approach to MSS, capable of handling long sequences with bidirectional processing, which enhances separation quality.
2. The authors employ both an objective metric (HAAQI) and perceptual feedback from users with hearing loss, which strengthens the validity of their results.

---

### Official Review · ~Zihan_Lv1 · 2024-11-11
**Sufficient research and innovative proposed method**

**Rating:** 8
**Confidence:** 4

**Review:**

The project has great potential to contribute to both MSS research and hearing aid technologies and conducts sufficient research. More experimental validation and discussion on the limitations might be needed given the controversial performance of Mamba.

---

### Official Review · ~Zhang_Mingkang1 · 2024-11-11
**Interesting topic**

**Rating:** 8
**Confidence:** 3

**Review:**

Strengths:

Background:

Compelling societal impact addressing hearing accessibility.
Well-defined problem scope combining ML and audio processing.


Definition :

HAAQI metric clearly defined with formula.
Good explanation of evaluation criteria.


Related Work :

Comprehensive review of audio source separation techniques.
Clear analysis of different architectural approaches (DNN, CNN, etc.).


Proposed Method:

Innovative use of Mamba architecture for audio processing.
Clear experimental configurations (specialized vs combined models).
Well-thought-out evaluation methodology.


Areas for Improvement:

Could provide more mathematical details about the Mamba architecture adaptation.
More specifics on handling real-time processing requirements.
Consider adding perceptual evaluation metrics beyond HAAQI.

---

### Official Review · ~Hector_Rodriguez_Rodriguez1 · 2024-11-11
**Review "Harmonic Clarity: Audio Source Separation Techniques on Classical Music"**

**Rating:** 10
**Confidence:** 4

**Review:**

The proposal presents a Mamba architecture for music source separation. The introduction provides a compelling case for the need to enhance hearing aids through music source separation. The related work highlights advancements in audio and music source separation, particularly addressing the challenges of separating instruments in small classical ensembles.

The proposed method and evaluation metric are clear and well established. The specialized and combined instrument approach leverages Mamba's strenghts and the evaluation metric is solid. The proposal could be further improved by specifying the data source that will be used for training and testing.

Overall, the proposal is clear, well-written, and meets all requirements.

---

### Official Review · ~Kaiyuan_Zhang6 · 2024-11-12
**Interesting yet need polish**

**Rating:** 7
**Confidence:** 3

**Review:**

An interesting topic focusing on audio source separation tasks for hearing aids, and well writtened. Some discussions on methods and evaluation have been proposed.
However, as a proposal paper, it is supposed to contain more expectations or future plans. For example, mentioning possible challenges and steps may be a good choice. Besides, more relevant research on the current amount of hearing aids patients need to be cited.
Overall, I would like to give it a 7 points.

---

### Official Review · ~Zhaoxi_Li2 · 2024-11-12
**Proposal Review: Harmonic Clarity – Audio Source Separation Techniques for Classical Music**

**Rating:** 8
**Confidence:** 3

**Review:**

This proposal presents a well-structured approach to advancing audio source separation (ASS) in classical music, a task with notable challenges due to the complexity and overlapping nature of musical elements in classical compositions. By adapting the Mamba state-space model, which excels at handling long-sequence dependencies and overlapping signals, the authors propose an innovative method that aligns with the needs of hearing aid users by enabling individualized instrument rebalancing. The approach is technically sound, particularly with its dual model configurations (specialized and combined models for instrument separation), which will help assess Mamba’s adaptability for ASS. The evaluation strategy is rigorous, combining the Hearing Aid Audio Quality Index (HAAQI) for objective analysis and perceptual tests for qualitative feedback. However, further details on comparative benchmarks and model optimization strategies would enhance clarity on expected outcomes and practical application potential. Overall, this proposal is promising, addressing a pressing need in audio accessibility with strong potential for impact.

---

### Official Review · ~Chumeng_Jiang1 · 2024-11-12
**Practically meaningful and thoroughly researched**

**Rating:** 8
**Confidence:** 4

**Review:**

This proposal focuses on developing advanced music source separation techniques, especially for classical music, aiming to improve accessibility for hearing-impaired listeners. The project plans to employ a state-space model architecture, Mamba, to enhance music source separation, exploring two specific approaches: specialized models for each instrument and a unified model for all instruments. Evaluation will involve objective metrics such as the Hearing Aid Audio Quality Index (HAAQI) and perceptual tests with hearing-impaired individuals.

Strengths:
- **Interesting and Significant research topic:** This research has real-world applications and can genuinely help people with hearing impairments.
- **Thorough related work:** The authors demonstrate a solid understanding of the field, having conducted detailed research on related work and identified its strengths and weaknesses.

Weaknesses:
- **Unclear methodology:** How will the specialized instrument models and combined instrument model be integrated?
- **Is the introduction of ASS necessary?:** Compared to ASS, MSS has a higher sample rate, but this doesn’t seem to be a critical feature in the proposed algorithm. Therefore, it doesn’t seem necessary to strongly differentiate between ASS and MSS.

---

### Official Review · ~Xuancheng_Li1 · 2024-11-12

**Rating:** 9
**Confidence:** 4

**Review:**

Summary
This proposal addresses the challenges of music perception for hearing-impaired individuals, particularly in classical music where complex compositions often overwhelm standard hearing aids. The authors aim to develop an end-to-end pipeline for Music Source Separation (MSS) that isolates and rebalances individual instruments, enabling the creation of personalized remixes to enhance clarity. This tailored approach could improve accessibility and the listening experience for users in both live and recorded music settings.

Strengths
The project tackles a unique and impactful problem, with a focus on emotional well-being and inclusion for the hearing-impaired. The proposal’s emphasis on rebalancing and remixing demonstrates a thoughtful approach to improving auditory accessibility and enhancing the richness of classical music for those with hearing challenges.

Weaknesses
The complexity of classical compositions may pose challenges for MSS accuracy, and details on specific techniques or models to address this are limited. Additional explanation of how the system will be evaluated in live vs. recorded settings would clarify its practical implementation.

Conclusion
This project offers an innovative approach to enhancing music accessibility for hearing-impaired listeners, with potential to significantly improve their musical experience. Further clarity on the MSS techniques and evaluation methods will be essential to assess the pipeline’s effectiveness in real-world applications.